# Phosphorus Recovery from Sewage Sludge Ash Based on Cradle-to-Cradle Approach—Mini-Review

Anna Jama-Rodzeńska [1], Józef Sowiński [1], Jacek A. Koziel [2] and Andrzej Białowiec [2,3,*]

1 Institute of Agroecology and Plant Production, Wrocław University of Environmental and Life Sciences, Grunwaldzki Square 24A, 50-363 Wrocław, Poland; anna.jama@upwr.edu.pl (A.J.-R.); jozef.sowinski@upwr.edu.pl (J.S.)

2 Department of Agricultural and Biosystems Engineering, Iowa State University, 605 Bissell Road, Ames, IA 50011, USA; koziel@iastate.edu

3 Department of Applied Bioeconomy, Wrocław University of Environmental and Life Sciences, 51-630 Wrocław, Poland

* Correspondence: andrzej.bialowiec@upwr.edu.pl

**Abstract:** The wastewater treatment process generates large amounts of P-rich organic waste (sewage sludge (SS)). The direct application of SS in agriculture, being controversial, is gradually being replaced by incineration, leading to the concentration of both P and heavy metals in the solid residual-sewage sludge ash (SSA). The novel closed-loop, cradle-to-cradle (C2C) approach leads to maintaining P production at current levels and counteracts its depletion in the future. The aim of this review is the presentation of the implementation of the C2C approach for P recovery. The paper focuses on steps that comprise P C2C, starting from the SS properties, being a derivative of wastewater type and treatment processes, to SS pre-treatment and finally leading to certified P-fertilizers production from SSA by application thermochemical or wet chemical extraction technologies. Examples of SSA treatment technologies and the final products are provided. It has been summarized that future research should focus on the production of SSA-based fertilizers aligning with the C2C concept and determining its effect on the various agriculture and horticulture crops.

**Keywords:** circular economy; wastewater treatment; resource recovery; sustainability; municipal waste; fertilizers; zero waste; thermochemical methods; wet chemical extraction; C2C





## 1. Introduction

The projected world population increase to ~10 billion by 2050 increases pressure on the food supply chain [1]. Phosphorus (P) is a critical resource, as it plays an essential role in food security [2]. Agriculture depends significantly on P [3,4], similarly to various industry sectors. The world's P demand includes the production of fertilizers (90%), feed (7%), food additives, fibre, fuel production and other industrial purposes [5,6]. The fertilizer use varies, and it is assumed that the P utilisation will grow by 70–100% in developing countries by 2050. The demand increase and the depletion of P resources require alternative P-sources and approaches [7,8].

Sustainable agricultural development with a future rock phosphate shortage should include two scenarios: reducing the P fertilizer load or increasing recycled P sources. The first scenario seems unlikely due to the growing population and their dietary requirements; therefore, the second is more likely.

Several factors contribute to the current uncertainty regarding the P stock, recovery and reuse. Standards for classifying P resources vary. Different data sources about P reserves and information are non-comparable across the literature [9]. According to different authors, the global P depletion will occur by 50–370 years [10–14].

The supply and location of P reserves are unevenly distributed throughout the world [15,16]. Most of the world's phosphate reserves are concentrated in Morocco, accounting for approximately 70% of the total deposit. Morocco is also the largest exporter

of phosphate rocks [17]. China is second in the ranking (6% of the world's phosphate reserves) [18]. Europe has small phosphate rock deposits and P demand depends strongly on its import from Morocco [19].

Phosphorus is considered as a "critical raw material"—a concept developed within the European Union (EU) Raw Minerals Initiative [4,20–22]. Due to that, P recycling and recovery options are considered part of a sustainable system [23]. Because of these challenges, steps should be taken to recover P on a larger scale from available waste materials based on the circular economy strategy and maintain its closed-loop cycle.

Figure S1 (Supplementary Material) shows the mind map of the P management system, including natural resources, bio-geo-cycles, the importance for living organisms and influence on eutrophication, extraction, application in agriculture for primary and secondary production, food processing, industrial production and discharge to the environment. The waste and wastewater processing industry has been derived as one sub-system, emphasizing a proper SS management hierarchy. This mind map indicates that raw sewage sludge (SS) and sewage sludge ash (SSA) derived from SS after different thermal treatment (incineration, pyrolysis, gasification) is significant P source [24]. SS management is emphasized because of the rapid increase of sludge production due to new wastewater treatment plants (WWTPs), sewage network extensions and upgrading these facilities [25]. The EU was expected to produce ~13 million tons of SS dry matter (DM) per annum by 2020 [24,26]. The annual global SSA production is ~1.7 million tons, which is expected to increase in the future [24,27]. SSA characterizes by a higher P content compared to SS. The proposed SSA use is supported by the estimation that ~80% of P is retained.

There has been limited experience with P recovery from wastewater combined with P reuse in agriculture. The presence of heavy metals (HM) in SS and SSA causes some challenges related to waste materials reuse and P extraction from SS and SSA. Therefore, the direct use of SS and SSA can be limited by HM content [24,26]. However, the HM content in SSA is 8 to 9 times higher than in SS (including toxic metals), usually above acceptable levels for fertilizer [28,29]. Additionally, without further processing of SSA, P is not plant-available due to bonding in complex compounds [30,31].

It is estimated that if struvite (STR) was recovered from WWTPs around the world, 0.63 million tons of P (as $P_2O_5$) could be recovered annually, reducing phosphate mining by 1.6% [32]. Struvite is the most readily recovered compound in pilot and operational plants in Europe, the USA, China and Japan [33]. Struvite production from an economic point of view is still unprofitable. However, the profitability of struvite production can become a reality [34] when P-fertilizers' prices rise. Struvite can be regarded as fertilizer with good quality because it contains two basic macroelements (N, P) and has relatively low solubility. The Mg content in struvite makes it an alternative fertilizer for some crops [35].

Most studies show comparable or higher efficacy of struvite compared to water-soluble fertilizers in the pot and field experiments. The results showed significant growth yield of plants fertilized with STR, e.g., Ricardo et al. (2009) [36] in lettuce cultivation, Wen et al. (2019) [37] in cabbage cultivation, Reza et al. (2019) [38] in Sudan grass, Bonvin et al. (2015) [39] in ryegrass (*Lolium multiflorum var. Gemini*) and Rhaman et al. (2014) [40] in grasses, vegetables, corn and fruits compared to the conventional water-soluble fertilizers. It can be explained that struvite is more effective than other commercial phosphate fertilizers due to the presence of Mg and the synergistic effect of the P/Mg ratio [41]. Fertilization with struvite also led to increased P content and uptake of P by plants [36,41,42]. Additionally, STR does not contaminate plants with HMs. Plants fertilized with STR have lower HMs concentrations compared to conventional fertilizers [37,43]. The closing of the P cycle and P recycling is a part of the cradle to cradle (C2C) approach concept. C2C-based goods production assumes recycling and reuse after their end of life [44,45]. C2C concept is a part of a "circular economy" where waste recycling and reuse are built into the process.

The research on C2C application and closing the P cycle by starting with waste and ending as a final product in agriculture is limited. This paper helps to develop a holistic

P cycling strategy adopting the circular economy concept in wastewater treatment. The investigated scenarios include the integrated P recycling from SSA and a waste-based P fertilizer as a final, safe and certified product in agriculture. While the possibility of applying the C2C approach within a closed-loop P cycle is promising, it requires further research and improvement, especially to implement these approaches at WWTPs and create policies and regulations encouraging and incentivizing it.

Recovered P in the form of struvite from WWTPs can be an answer to replace conventional fertilizers and help to reduce the long-term threat to food security posed by depleting phosphate resources. Phosphorus recovery under the C2C concept can provide opportunities for sustainable resource recovery from waste streams and protection of P reserves. Struvite can significantly contribute to solving selected agricultural and environmental problems (mitigation of eutrophication and 'dead' aquatic zones) and prevent costly problems for WWTPs [46].

This study aims to present P recycling from SSA according to the C2C approach. This work focuses on analyzing the P cycle starting in WWTPs (obtaining SSA through different methods) and ending with the final commercial product for application in agriculture and horticulture. The review aims to present step-by-step the whole process from WWTPs to the final product, highlighting critical points (e.g., HMs removal).

## 2. C2C Concept as a Sustainable Management Practice for P Fertilizer

The C2C strategy comprises three main principles:

(a)　waste is equivalent to food,
(b)　all wastes could be considered as nutrients for successive product life cycles and are treated as a part of biological and technological processes,
(c)　all wastes should be recycled for the subsequent processes [44].

For the C2C concept to be sustainable, all materials must be maintained at high purity, aiming for high efficiency in waste recycling. Therefore, the SS, coming from different sources, should be treated separately and management methods should ensure that SS does not burden the environment and allows potential future use in the circular economy [47–51]. This approach requires detailed and specific knowledge [52]. The P obtained from SS should return safely to the ecosystem at the end of the cycle. The P recycling from SS can have some drawbacks, especially if the recovered products have impurities. The presence of HMs in the SS may reduce the value of SS for P recovery. Only low HM SS should be considered for P recovery from SSA. Thus, when the quality requirements are maintained, the P derived from SS can be classified in C2C as a certified final fertilizer product [44,47,50,51,53,54].

Final products, including SS-based fertilizers obtained through the C2C approach, are defined as sustainable [55]. The C2C concept also includes functional alternatives and the needs of users who expect high-quality fertilizers (Figure 1).

The vital advantage for C2C is certification of the final product, which must meet certain requirements. Certification should be considered a reward for achieving satisfactory quality results [51], where products are assessed according to the following criteria waste health, waste reuse, renewable energy and carbon management, water management and social fairness [51].

The C2C approach allows for the integration of technology and science, which provides sustainable benefits. Apart from environmental aspects, it must be characterized by high production efficiency. The most important issue of this concept refers to the 'zero waste' approach and minimizing the adverse effects of toxic substances [51]. The C2C concept does not involve the depletion of waste resources (e.g., SS), except for the material used to construct the collection and storage facilities [44].

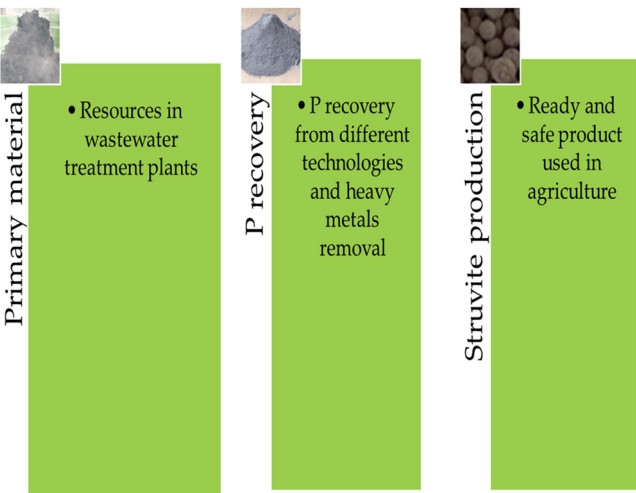

**Figure 1.** General diagram of struvite production.

## 3. Methodology

Challenges and opportunities in the introduction of struvite production were considered to propose the requirements for a sustainable nutrient management strategy (i.e., P recovery and recycling from SSA). The presented study mainly covers P recovery from SSA and marketed technologies allowing P recovery and HMs removal. The data collected and analyzed from the available literature allowed us to present an overall view of struvite formation based on the circular economy. The key aspect is the use of P-rich materials) through the removal of impurities so that the resulting product is safe (Figure 2).

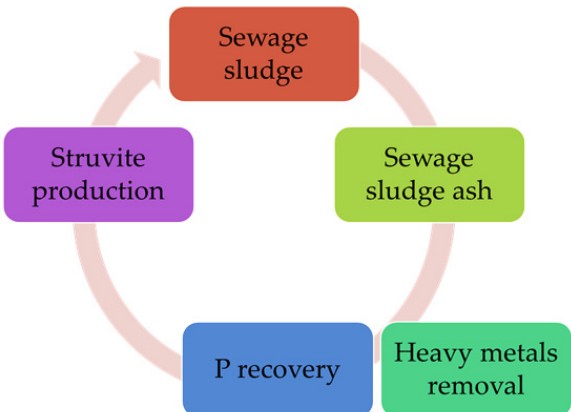

**Figure 2.** Scheme of struvite formation according to the circular economy concept.

## 4. Phosphorus Cycle and C2C Concept

The material flow analysis for the P recycling from SS and SSA is integral to transforming waste into a certified product for agriculture. The flow begins with P-rich SS, then follows with recycling treatment contributing to closing the P cycle with a certified product.

The research methodology of the closed cycle of P consist of five phases:

i. Characterization of SS and SSA waste with the potential associated with P production and demand.
ii. Determination of chemical composition SS and SSA waste, considering both the P and HMs content as the primary pollutants.
iii. Presentation of implemented technologies used for P recovery from SSA waste.
iv. Presentation of implemented technologies for HMs removal from SSA waste.
v. Production of sustainable and certified P-rich products recommended for agriculture.

The closed P cycle was discussed considering the four scopes of the C2C concept:

- Scope 1: Recycling of SSA with separation of P and HMs as a technological process.
- Scope 2: SS thermal treatment (incineration, pyrolysis) of SS for SSA generation as a pre-treatment process.
- Scope 3: P recovery from pre-treated waste with separation of HMs:
  - Sub-scope 3a: SSA recycling for P recovery as a substrate to new products,
  - Sub-scope 3b: Enrichment of the purity and elimination of HM content.
- Scope 4: Development of certified products.

The most important is scope 1 as being essential for scope 4 and linked with the production of SS with different nutrient content without undesirable substances (Sub-scope 3a). Scope 2 comprises the transformation of SS to SSA. Incineration of SS is one of the options to generate SSA that is considered highly efficient, environmentally friendly and economical. The need for HMs removal from heavily polluted SS was discussed at sub-scope 3b. Scope 4 comprises the processes of technology aiming to produce final, certified products.

The technical principles of P recovery technologies have often been published, but the information on distinguishing materials with highlighting closed P cycle is missing. The previous work of Egle et al. [56,57], Amann et al. [58], Poluszyńka et al. [59], Kasprzyk and Gajewska [60] and others focused on the basic technical background and detailed material flows, processes and technologies of P recycling. However, holistic combining of these elements as a closed circuit into one cycle is missing. This gap in the puzzle of P circulation is addressed herein.

## 5. Sewage Sludge Ash—The Material Flow in the C2C Approach

### 5.1. Sewage Sludge Characterization as a Source of P and Heavy Metals, before Thermal Pre-Treatment in the C2C Concept

SS can be characterized by various physical, chemical and microbiological parameters and consists of a wide range of water 55–99% (99%-raw sludge, 55–80%-dehydrated sludge). The varied sludge composition from different WWTPs creates the need for local solutions targeting specific types of treated wastewater, technologies (mechanical, biological, chemical) and the problem of SS disposal [3,28,61–63].

Council Directive 86/278/EEC on the protection of the environment, particularly the soil, when SS is used in agriculture) [64] provides the exact definitions of SS in Article 2. The SS disposal in agriculture is often chosen due to the nutrient value of sludge determined by N, P, Mg and Ca concentration. Nitrogen and P are the most valuable nutrients in SS (Table 1). Other water-soluble nutrients (K, Ca and Mg) are present in lesser amounts [65,66] (Table 1).

**Table 1.** Examples of the chemical composition of sewage sludge [67–71].

| Chemical Composition | Units | The Amount of Element in Sewage Sludge |
| --- | --- | --- |
| Acidity/basicity | pH | 6.64–8.83 |
| Dry mass | % | 16.60–41.03 |
| Organic matter | g kg$^{-1}$ DM | 50.60–55.83 |
| Total nitrogen | g kg$^{-1}$ DM | 22.40–38.90 |
| Nitrate nitrogen | g kg$^{-1}$ DM | 3.60–32.00 |
| Total phosphorus | g kg$^{-1}$ DM | 5.00–25.00 |
| Potassium | g kg$^{-1}$ DM | 2.34 |
| Calcium | g kg$^{-1}$ DM | 2.80–16.60 |
| Magnesium | g kg$^{-1}$ DM | 6.10–8.00 |
| Lead | g kg$^{-1}$ DM | 0.11–0.07 |
| Cadmium | g kg$^{-1}$ DM | 0.001–0.002 |
| Nickel | g kg$^{-1}$ DM | 0.016–0.028 |
| Zinc | g kg$^{-1}$ DM | 0.70–1.29 |
| Copper | g kg$^{-1}$ DM | 0.126–0.130 |

Phosphorus occurs in SS in various forms: organic and inorganic, the latter being more plant-available [8]. The total P content of SS can be varied from 5 to 25 g kg$^{-1}$ DM [67]. In the case of fertilizer production, the most valuable information is the content of the biologically available P in the SS, determined as the sum of the available P including NaOH-P and HCl (1M)-P and HCl (3.5M)-P fractions [67]. Phosphorus also occurs in combination with Al or as a component of organic compounds, while in combination with Cd is negligible [67]. About 10% of total P in SS is present in water-soluble and easily extractable forms of phosphates [65,72,73]. The composition of inorganic P depends on the chemicals used in the wastewater treatment process for P removal and improving the sludge dewatering [74]. About 67–69% of P in SS is concentrated in fly ash after the SS incineration process. In contrast, all of the organic P is volatilized [75].

The presence of HM is a restricting factor towards the use of sludge for agricultural purposes (86/278/EEC) [64]. In addition, the HMs, SS contains certain groups of organic compounds, including persistent organic pollutants (POPs) [76]; thus, their fate should also be tracked and considered for the proposed P recovery and HM removal technologies. SS is potentially hazardous because it may contain Cr, Cd, Se, As, Co, Pb and Hg. The level of HM content in SS ranges from <1 mg·L$^{-1}$ to 1000 mg·L$^{-1}$. On a dry weight basis, the HM concentrations range from 5–20 g kg$^{-1}$; however, in some cases was reported to be as high as 60 g kg$^{-1}$ [77]. During the SS incineration, due to the oxidation of the organic fraction of solids, the content of HM increases. Untreated SSA contains As, Cd, Cu, Ni, Pb and Zn at levels exceeding threshold values for application in agriculture [77].

*5.2. Sewage Sludge Ash Characterization as a Source of Phosphorus and Heavy Metals, after Thermal Pre-Treatment of SS in the C2C Approach*

Chemical composition, type of wastewater, treatment and processing technology impact SSA quality [78]. SSA contains P (~130 g·kg$^{-1}$ DM), Ca (~140 g·kg$^{-1}$ DM), Mg (~30 g kg$^{-1}$ DM) and K (~10 g·kg$^{-1}$ DM).Ashes from SS incineration contain 40–90 g kg$^{-1}$ DM of P [12]. Kruger et al. [30] and Herzel et al. [79] reported that SSA has a high P content ranged from 15 to 131 g kg$^{-1}$ DM. Cieślik et al. (2015) [80] determined that ash from SS incineration may contain P from 70 to 134 g kg$^{-1}$ DM. This higher P content is comparable to apatite (120–160 g·kg$^{-1}$ DM) [65,66]. Adam et al. (2009) [26] reported the content of P$_2$O$_5$ from ash being even higher (140 to 250 g·kg$^{-1}$ DM).

SSA may contain sand-sized particles with low moisture and negligible residual organic matter [81]. The main compounds are silicon oxides (166 g kg$^{-1}$ DM), aluminum oxide (51 g kg$^{-1}$ DM) and calcium oxide (129 g kg$^{-1}$ DM). SSA is also characterized by the presence of quartz, hematite and anhydrite [82]. Quartz and hematite constitute the most abundant minerals in SSA, while other (Fe, Ca and Al phosphates) are also important [83].

HM in SSA occur in different forms, primarily oxides and hydroxides [84]. Zn, Cu, Cr, Pb, Ni and Cd dominate. However, these elements were only present in a small fraction of SSA samples [83]. Significant variation in HM concentrations in SSA may result from differences in wastewater treatment systems, incineration conditions and method of testing [85]. Białowiec et al. (2009) [86] compared two types of SSA coming from SS incineration plant in Vienna (Austria) and Gdynia (Poland). They revealed that both ashes were characterized by high mineralization; the organic C content ranged from 1.3 to 1.6 g kg$^{-1}$ DM. The ash was low in N (0.11 g kg$^{-1}$ DM). The P content differed significantly, i.e., exceeded 50 g kg$^{-1}$ DM (Vienna), while lower in Gdynia (19 g kg$^{-1}$ DM). Analyses of HM content showed differences, although their concentrations and mutual proportions resulted from the typical characteristics of SS. Gdynia ash was characterized by Zn content 3×> than ash from Vienna (7762 mg kg$^{-1}$ DM). In the case of other metals, the relationship was the opposite; Viennese ash contained more HMs, especially Cr (~20× more (54.5 mg kg$^{-1}$ DM) and Hg (~12× more, but in the low concentration range, 0.074 mg kg$^{-1}$ DM) and an increased Cu concentration (703 mg · kg$^{-1}$ DM). Białowiec et al. (2009) [86] reported that the chemical composition of water extracts showed relatively low susceptibility to washing out pollutants from the ashes. The obtained extracts were characterized by a low content of dissolved substances ranging from 2.5–3.8 g kg$^{-1}$ DM. There was a tendency to alkalize

the water environment, with Viennese ash increasing the pH up to 11.29 and Gdynia ash to 8.55, which resulted in the very low leachability of HM.

## 6. Heavy Metals as a Determinant Factor of SSA Technical Process in Cradle-to-Cradle Approach

Removing HM from SSA is crucial because they are not degradable and pollute soil, crops and water [87]. The most critical issue of fertilizer safety is the concentrations of HMs below accepted limits and the availability of P from the product derived from waste [88]. The best option would be to achieve a low HM sludge content; however, it is not always possible or practical. The extraction of HM from sludge before further use is necessary to ensure the high environmental standards of SSA-based fertilizers.

Heavy metals constitute a technical challenge in the production of fertilizers; therefore, they should be addressed. There are many methods of removing HMs from ash, but thermal and thermochemical methods are considered as most effective. The volatilization of HMs depends on many factors, such as time, temperature, metal grade, matrix type, characteristics of waste, type of incinerator, system configuration, air introduction techniques and use of the catalysts for metals volatilization [89,90].

Thermochemical methods are regarded as the most effective process in HMs removal from SSA below the accepted limits for fertilizers and increase the P bioavailability. Thermochemical processes separate HM and P based on volatilization processes at high temperatures. SSA is heated (depending on the method from 500 to 1000 °C [91]) in the reactors that achieve high HM removal rates. These conditions lead to the formation and evaporation of volatile HM compounds (mainly chlorides) and the production of less mobile metal forms [92]. About 90–99% of HMs such as Cd, Cu, Pb and Zn, can be removed using thermochemical methods and the solid product after the treatment can be directly used to produce fertilizers.

## 7. Sewage Sludge to Phosphorus by Thermochemical Processes

P recovery means the recapture of this element from SS via SSA. This process can be conducted by following different methods: wet chemical extraction (e.g., Sephos process, Biocon, Pash process, Eberhard process), high-temperature processes (e.g., Merphec, Susan, ATZ ion bath reactor), bioleaching process (Incore) [93–95]. P from SSA can be recovered using thermochemical processes, which allow the transformation of ash into marketable fertilizer products according to the circular economy and C2C approach [79].

The first step is converting SS into SSA directly in incineration or co-incineration, or indirectly via SS pyrolysis before the incineration. It is constricted with solid residuals and significant differences in their composition and phosphate fertilizer values. These processes differ not only on the type of the final products, quality for agricultural use and, more importantly, their impact on the environment.

The main product in the SS incineration is fly ash [96,97]. Incineration temperature influences the ash's properties leading to a lower P concentration in the ash fraction and greater in the volatile fraction. The combustion destroys hazardous organic compounds in the sludge and reduces unpleasant odors [24]. Dust fraction, in turn, contains contaminants captured due to the mandatory exhaust gas treatment [8]. Ash after combustion could be used as a fertilizer, however, not directly but after suitable treatment [98].

If SS is not suitable as fertilizer, it should be incinerated. Incineration is considered to be the best available technology (BAT) [99] for SS disposal at a high temperature which can reduce the volume of material [79,100]. During incineration, P takes the form of volatile oxides, condensing upon cooling to a temperature of 400–600 °C to form $P_4O_{10}$, which can clog the filters [80,101]. However, the SS incineration is not a waste-free method because ash still contains HM and cannot be used directly in agriculture as this would pollute soil and impact living organisms [102].

Incineration has become a popular method of SS management in Poland and Europe. Thermal treatment is not neutral for the environment; therefore, it constitutes the critical point of C2C design to be considered sustainable. There are many problems connected

with incineration [103]. There is considerable public concern about exhaust emissions and atmospheric pollution. Still, the amount of sludge utilized thermally has increased in Poland from 1.6% to 18% [104,105]. Thermal waste treatment technology became an integral part of the municipal waste management system in large cities and urban areas [106]. Dominant technologies are based on fluidized bed combustion. The capacity of such installations ranged from 64 Mg d$^{-1}$ to ~70,000 Mg y$^{-1}$ [106]. Approx. 11% of sludge (57,000 Mg DM in 2012) was thermally treated compared to 3.7% in 2010 [107]. The processing capacity was 160,000 t d.s. y$^{-1}$ in 2018 [105].

When incineration/combustion conditions are not ideal, they can cause inefficiency, mainly due to inconsistent temperatures and mixing conditions inside the furnace [108]. These inefficiencies may cause the formation of toxic incomplete combustion products. PAHs are of particular concern because of their relative abundance in emissions and impact on human health [108].

Gases emitted during incineration also comprise greenhouse $CO_2$ (including biobased), $CH_4$ and $N_2O$, measured as $CO_2$ equivalents ($CO_2$eq). Tarpani et al. (2020) [109] determined the environmental impact of SS incineration. SS incineration results in the low impact of climate change potential considering the value of 39.8 kg $CO_2$eq t$^{-1}$ DM; however, this may contribute to ozone depletion, eutrophication and acidification potential. Thus, SS incineration leading to P recovery may still place a burden on the environment. Therefore, ecotoxicological assessments should be considered [110].

## 8. Phosphorus Recovery from SSA

### 8.1. Available Methods of Phosphorus Recovery from SSA

The central point of P circulation in C2C is its recovery in a complex process that requires appropriate installation, chemicals and staff training. Because much of the work on the P recovery concerns these points, the following narrative has been reduced to a few most effective methods. In addition, current options of P recovery, other technologies are under development and testing, while only a few technologies have been introduced in recent years.

Two primary ways of processing ash from SS incineration are wet chemical extraction and thermochemical methods. These approaches differ by the origin of the used matter (wastewater, sludge slurry, fermented or nonfermented sludge ash) and the process (precipitation, wet-chemical extraction, thermal treatment).

Thermochemical methods are included in the later process of P production in an electric furnace and ash calcination. The plant availability of P in SSA depended on the thermal conversion process, causing a significant change in the molecular environment of phosphates compared to the original sludge [111]. During these processes, P is transferred into a mineral form available for plants [96]. However, 18% of the P contained in the fertilizer remains in an unavailable form [112]. Obtaining P in a solid phase constitutes the basis for forming new phosphate compounds with high plant availability [2,113].

### 8.2. Available Technologies of Phosphorus Recovery from SSA

8.2.1. The Technologies of Wet Chemical Extraction of P from SSA

The Biocon process was developed in Denmark by PM Energi A/S (Neminnen 2010). The BioCon® (Phosphorus recovery process) (Table 2) process relies on P recovery from thermally treated SSA at 850 °C. The process leads to receiving phosphoric acid from SSA. The process relies on the P and HMs dissolving. SSA is subjected to sulfuric acid until pH 1.0 is attained. Then, the solution is subjected to go through a sequence of ion exchangers. The final exchanger collects phosphates. The process uses an ion exchange that allows the purification of orthophosphoric acid and available forms of phosphates. The ash is leached with acid and the content in the leachate is separated with ion-exchange technology [114].

Sequential precipitation of P (SEPHOS) has been developed at the Technical University of Darmstadt (Table 2). This process is based on the wet chemical methods of P recovery from SSA, which is treated with sulfuric acid at a pH < 1.5 to dissolve P and HMs. Phosphate

is produced with high Al content. SEPHOS achieves 12% P content, compared to 9.8% in the ash before treatment and a significantly reduced HM content [115].

**Table 2.** The wet chemical extraction methods of phosphorus from SSA [31,56,60,116,117].

| Technology | Equipment | Chemicals | Final Product |
|---|---|---|---|
| BioCon | reactor | $H_2SO_4$ | phosphoric acid |
| SEPHOS | reactor | $H_2SO_4$, NaOH | calcium phosphate (12% P) |
| PASH | Centrifuge, equalizing tank | HCl (8%), NaOH, $H_3PO_3$, $H_2SO_4$ | phosphorus potential 90% |

P recovery from ash (PASH) process has been developed in the Aachen University [114] (Table 2). The PASH is an example of a wet chemical method showing 70–80% recovery rates relative to the ash input. The highest solubility is achieved using HCl (8%). Reduction in the HM concentrations reaches 95%. It is a "mono-incineration" process and calcium phosphate or struvite is the final product [56,58].

### 8.2.2. The Technologies of Thermochemical Recovery of P from SSA

Another approach to P recovery is RecoPhos®, an example of a thermo-chemical process that comprises the fractioned extraction of phosphate and HMs from SSA mixing with phosphoric acid at high temperatures [56,118] (Table 3). RecoPhos® technology produces elementary P. It facilitates the thermochemical reduction of phosphates at >1500 °C and evaporation of elementary P. The process results in a high-quality fertilizer with 16.6% P content [56].

ASHDEC is destined as mono-incinerated SSA by treating it with chloride and thermal treatment to remove HMs (Table 3). The final product is suitable for agricultural use. The ash is mixed with $CaCl_2$, KCl and $MgCl_2$, then the fillers are added. The mixture is pelletized or granulated. The granules are heated at 1000 °C for up to 20 min. The first installation in Leoben (Austria) in 2009 processed 7–10 t of ash per day [60]. ASH DEC WWTP can produce about 51–85 kt y$^{-1}$ of a P-rich product with a $P_2O_5$ content from 13 to 18% [119]. The ASH DEC technology reported excellent HM removal rates and high P recovery rates of ~98%. The final product was licensed under the name PhosKraft as a high-quality fertilizer [119]. The advantages of ASH DEC technology are the relatively low cost of P recovery from ashes, high enrichment (approx. 20% $P_2O_5$) and compliance with the EU requirements for the production of artificial fertilizers [60].

In the "Wöhler process," the most critical operation is that phosphates react with carbon and silicon dioxide in a furnace and are reduced to elemental P [120] (Table 3). The InduCarb retort is used, where a coke bed is heated inductively. Reduction of the P contained in the SSA takes place in a thin melt film on the surface of the coke particles [88,118,121].

The metallurgical P and energy recovery from dry SS (MEPHREC®) installation is at a pilot stage (Table 3). The advantage is its potential application with various types of waste containing P. Dried SS is briquetted with slag-forming substances. The mixture is treated at 2000 °C and slag is enriched with P [56,60]. The final granulate is a highly digestible fertilizer with a high P content. While the potential for P recovery is calculated at 60–80% [56,60], the MEPHREC® is characterized by a low HM removal rate [56].

Slivberwerking Noord-Brabant (SNB) uses ash from the incineration of SS, allowing P recovery. Sludge produced is first dried to a water content of 60% and then incinerated at 850–950 °C in a fluidized bed. Ash produced contains ~80 g of P kg$^{-1}$ ash. The ash is then marketed to P recovery companies [60]. However, this method is regarded as high energy-consuming and not economical [60].

**Table 3.** Thermochemical methods of phosphorus recovery from SSA [31,56,60,116,117].

| Technology | Equipment | Time of Incineration | Temperature | Other Reagents | Final Product |
|---|---|---|---|---|---|
| ASH DEC | mono incineration of sewage sludge | 20 min | Up 1050 °C | - | 20% $P_2O_5$ |
| Slivberwerking Noord-Brabant (SNB) | fluidised bed boilers | - | 850–950 °C | - | ash contains in 1 kg–80 g of P |
| RecoPhos | mono incineration of sewage sludge; reactor InduCarb, (inductively heated) | - | 1900 °C | $H_2PO_4$ | $P_2O_5$–22% |
| Bama/AshDac | Thermally treated Silo, rotary kill, mill | 60 min 20 min (1000 °C) | 850–1000 °C | Chlorine donor | $P_2O_5$ content of 13–18%, |
| Merphec | Thermal volarisation; Furnace heat exchanger boiler plant steam turbine; gas scrubber | - | 2000 °C | - | 4.6–12% $P_2O_5$ with over 90% citric acid solubility |

P recovery from ash produced from SS incineration is still unresolved in Poland, both in organizational and legal aspects. Despite the many reasons for dealing with P recovery from ash, no regulations have been introduced in Poland. However, the produced ash is either disposed of at the plant's landfill for non-hazardous waste or passed onto external companies. The cost of such installations (SS mono-combustion plants) is ~6x higher compared with agricultural use [107]. Despite the increasing use of thermal installations for SS incineration to obtain ash, the technology of P recovery from ash is still in the testing phase. To this day, there is no WWTP in Poland recovers P from ashes and produces certified fertilizer. A limited number of WWTPs in Poland are recovering P from SS and producing fertilizer from it.

**9. Fertilizers Made from SSA Closing the C2C Loop and the System of Certification**

Certification procedures characterize the C2C compared to other systems aiming to support companies that produce according to C2C approaches. The C2C Certified Products Standard (now in its third version) is based on the C2C design principles established by William McDonough and Michael Braungart [51]. The following criteria are considered in the C2C certification: material health, material reuse, renewable energy, water stewardship and social fairness [51]. Companies can obtain a certification mark for their C2C product. This mark, which expresses the level of certification, can be placed on the product [122]. Considering material health, the ABC-X assessment methodology is applied to classify materials based on the chemical risk and its recyclability. In this category, it is possible to obtain A, B, C, X, or Grey (unknown) score. For each category, a product is assigned an achievement level (basic, bronze, silver, gold, platinum). The lowest achievement level of a product in a given grey category is identical to the certification level; the highest level is platinum. C2C certification is valid for 2 y; after that period, companies must again recertify. It concerns all products currently C2C-certified, although companies can begin the recertification process following V4.0 [123].

P obtained from SSA is the final stage of the C2C concept. Production of P fertilizers closes the nutrient cycle and starts a new one that could be applied to the soil and enhancing plant productivity and soil fertility. Some technologies implemented and used SSA for fertilizer production are listed in Table 4. Various technologies are developed to enrich the nutrients of SSA to use it for sustainable fertilizer as a convenient granular form of

fertilizers with low HMs and high P content. The form of the final product is essential to ease transport, storage, application and the uniform release of nutrients. Therefore, granular fertilizers seem to be appropriate [74,112]. Many fertilizers produced based on SSA have been tested regarding their agronomic suitability and impact on plants' yield and growth [79,118]. Some of them can be used as complete fertilizer to meet the nutritional requirements of plants.

**Table 4.** Fertilizers produced on the base of SSA as a product ending the closed phosphorus cycle.

| Name of the Product | Material Used to Its Production | Composition | Country | Reference |
|---|---|---|---|---|
| SSAB (SS ash–Bacillus) | Ash from SS | P-solubilizing microbes (PSM): Bacillus megaterium | Poland | [124] |
| Gifu-no–Daichi® | SSA | Granulated fertilizer or sold as soil improver | Japan | [125] |
| PhosCraft | SSA | NPK 20-8-8 | Austria Germany | [119] |
| PolFerAsh | SSA | - | Poland | [74] |

Another critical point of the C2C concept is the lack of standard testing of such fertilizers. Sludge management faces a significant task in producing a safe fertilizer that farmers will adopt. Farmers might be wary of using SS-derived fertilizer without certification. An example of such fertilizer is Gifu-no-Daichi®, a fertilizer that is marketed through the JA-Zen-Noh (National Federation of Agricultural Cooperative Associations) in Gifu City with a favorable reputation from farmers [125]. Most farmers in Palestine supported the concept of using SS and were aware of the properties, advantages and disadvantages of using sludge in agriculture. A small percentage of farmers did not support the use of SS due to psychological and social concerns, potential health risks or religious beliefs [126].

A larger percentage of farmers will accept the application of SS on agricultural land if consumers buy agricultural products fertilized with sludge. At the same time, SS should fulfill the social requirements and not exceed the market prices of conventional fertilizers. Good quality, safe to handle, lack of contaminants, environmentally friendly and competitive pricing can be attractive for farmers. The depletion of phosphate fertilizers and their high prices could additionally encourage farmers to use ash-based fertilizers.

The cost of P recovery from wastewater is still several-fold higher than the price of fertilizer production from rock phosphate [127]. The financial cost of struvite production should consider construction costs, operating costs and potential benefits to be found from the reuse and sale of struvite [128]. The costs of reactors and separation units for struvite precipitation are quite low. The greatest influence on the cost of struvite extraction is the operating costs, which depend on the type and characteristics of the wastewater, especially the addition of reactants [129]. For P recovery, the use of Mg reactants can comprise up to 75% of the total costs [130]. The average selling price of struvite is 0.41 €/kg, the use of $MgSO_4$ proved very expensive to generate a profit [131]. It has been found that an inexpensive source of Mg can reduce costs by 18–81% [132] as seawater concentrate (nanofiltration, reverse osmosis). According to Ueno and Fujii (2001) [133], the struvite can be sold from the sewage dewatering sludge treatment at 245 EUR/ton. Based on experiments, in full-scale WWTPs in Japan, the income from struvite is only about one-third of the cost of chemicals alone [134].

## 10. Conclusions

Reducing reliance on phosphate rock can be addressed by recovering and reusing P in agriculture from sewage sludge ash (SSA). Phosphorus recovery from SSA based on the cradle-to-cradle (C2C) concept appears to be rational and technologically feasible. This is considering the operating technologies and installations, some with advanced technologies, aiming to obtain high purity, certified fertilizer for use in agriculture. C2C concept will

save the P deposits, contribute to implementing zero waste and make food production systems more sustainable. C2C closes the loop and creates more sustainable products that can be marketed based on the established certification system. Several mature technologies are operational and ready for broader adoption when the socioeconomic conditions are favorable. The concept of circular economy based on C2C reuse, recycling and imitation of natural cycles in practice could be introduced on large scales and other critical minerals and resources. There is still a large gap between theory and practice that needs to be gradually implemented and this gap should be completed.

**Supplementary Materials:** The following are available online at https://www.mdpi.com/article/10.3390/min11090985/s1, Figure S1: The mind map of the P management system where the sewage sludge management hierarchy is an essential part of the Cradle-to-Cradle concept.

**Author Contributions:** Conceptualization, A.J.-R. and J.S.; methodology, A.J.-R., J.S., A.B.; software, J.A.K.; validation, J.S., A.J.-R. and A.B.; formal analysis, J.A.K.; investigation, A.J.-R.; resources, J.S.; data curation, A.J.-R., J.S.; writing—original draft preparation, A.J.-R., J.S.; writing—review and editing, A.B., J.A.K.; visualization, A.J.-R.; supervision, J.S., A.B., J.A.K.; project administration, A.B.; funding acquisition, A.J.-R. All authors have read and agreed to the published version of the manuscript.

**Funding:** This research was funded by the "Innovative scientist" program (grant number B030/00110) from the subsidy increased for the period 2020–2025 in the amount of 2% of the subsidy referred to Art. 387 (3) of the Law of 20 July 2018 on Higher Education and Science, obtained in 2019. The publication is financed under the Leading Research Groups support project from the subsidy increased for the period 2020–2025 in the amount of 2% of the subsidy referred to Art. 387 (3) of the Law of 20 July 2018 on Higher Education and Science, obtained in 2019. Jacek Koziel's participation was partially supported by the Iowa Agriculture and Home Economics Experiment Station, Ames, Iowa. Project no. IOW05556 (Future Challenges in Animal Production Systems: Seeking Solutions through Focused Facilitation) sponsored by Hatch Act & State of Iowa funds.

**Acknowledgments:** The presented article results were obtained as part of the activity of the leading research team—Waste and Biomass Valorization Group (WBVG).

**Conflicts of Interest:** The authors declare no conflict of interest. The funders had no role in the design of the study; in the collection, analyses, or interpretation of data; in the writing of the manuscript; or in the decision to publish the results.

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
