# Peer review of "Phosphorus Recovery from Sewage Sludge Ash Based on Cradle-to-Cradle Approach—Mini-Review"

_minerals, doi:10.3390/min11090985_

Round 1

Reviewer 1 Report

The paper presents an interesting and trendy topic, which will likely be interesting to various readers. However, because this topic is very interesting, there are a lot of research articles that are examining it. I would recommend finding some of them and expaining their research in the Introduction segment. The explanation of C2C in the Introduction segment would be better suited as its own chapter.

Another small comment - it's not necessary to describe the articles in Directives - it would be enough only to mention them, where readers can follow them themselves.

Moreover, I would recommend removing the metodology concept in the Methodology chapter. The reader gets the idea that some sort of methodology will be developed, while the paper provides an overview of technologies and methods on P recovery. Methodology chapter could provide an overview how the research and literature overview was conducted and what were the ideas behind it.

Author Response

Out responses to the Reviewer's comments are in the attached file.

Reviewer 2 Report

The manuscript entitled "Phosphorus recovery from sewage sludge ash based on cradle-to-cradle approach – mini-review", revises some interesting points about the recovery of phosphorus from sewage sludge, but in my opinion, needs extensive English editing and it would be useful to integrate or develop diagrams that would explain the purification processes of phosphorus or even the energetic expenditure or operation costs which these processes rely on.

Author Response

(The authors gave the same response as above.)

Reviewer 3 Report

In the article, the authors discuss an important issue related to phosphorus recovery from the sewage sludge ash. The work is interesting and extensively described. However, there are some elements that need to be completed in the manuscript. The novelty aspect is weakly emphasized, the goal also needs to be improved. The work is review and therefore I miss conceptual diagrams. The work contains only 4 tables. The weakness of the work is the lack of drawings, diagrams that would be an interesting supplement to the text. 

Author Response

(The authors gave the same response as above.)

Round 2

Reviewer 2 Report

The manuscript entitled "Phosphorus recovery from sewage sludge ash based on cradle-to-cradle approach – mini-review", was subjected to an extensive review, which the authors considered the suggestions of the revisors to enhance the quality of this work. In my opinion, this manuscript is ready for editorial revision and publishing.